# PI3K/AKT/mTOR Signaling Pathway in HPV-Driven Head and Neck Carcinogenesis: Therapeutic Implications

**DOI:** 10.3390/biology12050672

**Published:** 2023-04-29

**Authors:** Francisco Aguayo, Francisco Perez-Dominguez, Julio C. Osorio, Carolina Oliva, Gloria M. Calaf

**Affiliations:** 1Departamento de Biomedicina, Facultad de Medicina, Universidad de Tarapacá, Arica 1000000, Chile; 2Laboratorio de Oncovirología, Programa de Virología, Instituto de Ciencias Biomédicas (ICBM), Facultad de Medicina, Universidad de Chile, Santiago 8380000, Chile; 3Instituto de Alta Investigación, Universidad de Tarapacá, Arica 1000000, Chile

**Keywords:** PI3K/Akt, cancer, oropharyngeal

## Abstract

**Simple Summary:**

A subgroup of cancers that arises in the zone of the head and neck (HNCs) are caused by some types of human papillomavirus (HPV), so called high-risk (HR)-HPVs. HR-HPVs promote signaling pathways alterations involved in the initiation and progression of cancer. Among them, phosphatidyl inositol 3-kinase (PI3K)/AKT/mTOR signaling is involved in increasing cell proliferation, migration, and invasion. In this review, we dissect the role of PI3K/AKT/mTOR in HR-HPV-associated HNCs development and the impact as a potential therapeutic target.

**Abstract:**

High-risk human papillomaviruses (HR-HPVs) are the causal agents of cervical, anogenital and a subset of head and neck carcinomas (HNCs). Indeed, oropharyngeal cancers are a type of HNC highly associated with HR-HPV infections and constitute a specific clinical entity. The oncogenic mechanism of HR-HPV involves E6/E7 oncoprotein overexpression for promoting cell immortalization and transformation, through the downregulation of p53 and pRB tumor suppressor proteins, among other cellular targets. Additionally, E6/E7 proteins are involved in promoting PI3K/AKT/mTOR signaling pathway alterations. In this review, we address the relationship between HR-HPV and PI3K/AKT/mTOR signaling pathway activation in HNC with an emphasis on its therapeutic importance.

## 1. Introduction

Head and neck cancers (HNCs) harbor a group of heterogeneous tumors arising in the zone of the head and neck, including in the lips, oral cavity, nasal cavity, paranasal sinuses, oropharynx, larynx, and nasopharynx [1]. Head and neck squamous cell carcinomas (HNSCCs) are the most common histological type of HNCs, encompassing more than 90% of all diagnosed tumors [1]. HNCs were the sixth leading cancer and the seventh cause of cancer-related deaths in 2020, with 931,931 new cases and 467,125 deaths worldwide, respectively [2]. Among them, oral, larynx and nasopharynx cancers were the most frequent, with 377,713, 184,615, and 133,354 new cases in the same year, respectively [2]. Even though decreased incidence rates were reported for laryngeal and nasopharyngeal carcinomas from 1990 to 2017, the frequency of lip and oral cavity cancers, as well as other pharyngeal tumors, increased worldwide (reviewed in [3]). The incidence rates of HNCs showed geographical variations. For instance, lip and oral cavity tumors were most frequent in Melanesia, South Central Asia, Eastern Europe and Australia/New Zealand and they were less common in countries in Africa and Central America [2]. Tobacco smoking and alcohol consumption are well established risk factors involved in HNC development [4]. Additionally, viral infections are considered risk factors associated with the development of subsets of HNCs. Indeed, undifferentiated NPCs are consistently associated with Epstein–Barr virus (EBV) infection [5], while infection with specific types of human papillomavirus (HPV), so-called high-risk (HR)-HPVs, have been etiologically associated with a subset of oropharyngeal cancers, HPV16 being the most prevalent genotype [6,7]. Clinical data revealed that HR-HPV-driven oropharyngeal carcinomas are a particular clinical entity with an improved outcome, survival, and therapy response [8]. The molecular mechanisms involved in such clinical behavior have not been fully elucidated. The phosphatidylinositol (PI)-3-kinase (PI3K)/Akt/mTOR signaling pathway is associated with cancer progression, angiogenesis, chemotaxis and invasiveness [9]. Frequent gain-of-function mutations in PI3K gene have been found in solid tumors; furthermore, this pathway can be activated by oncoproteins and/or mutated tyrosine kinases [10]. The PI3K/AKT/mTOR pathway is hyperactive in ~90% of HNSCCs due to epidermal growth factor receptor (EGFR) activation (47%), PTEN mutation (10–15%), PIK3CA mutations (8.6%), PIK3CA amplifications (14.2%), and PI3K overexpression (27.2%) [11]. Furthermore, PI3K/AKT/mTOR hyperactivation is frequently found during radiotherapy and cytostatic drug resistance [11]. Importantly, dysregulations in the PI3K pathway are common in both human papillomavirus (HPV)-positive and -negative HNSCCs [12]. Indeed, viral oncoproteins such as HPV E6 and E7 can deregulate EGFR/PI3K/Akt/mTOR and increase PIK3CA gene mutations [13]. In this review, we address the mechanisms by which PI3K/Akt/mTOR is activated in HPV-driven HNC and its importance as a target for chemotherapeutic agents.

## 2. PI3K/Akt/mTOR Activation

### 2.1. PI3K/Akt/mTOR Activation under Physiological Conditions

The PI3K/Akt/mTOR pathway is critical for cell growth and survival, maintaining cell homeostasis under physiological conditions [14]. PI3K is a membrane enzyme that phosphorylates the D3 position of the inositol ring of PI(4,5)P2 to produce PI(3,4,5)P3 in response to extracellular agonists such as growth factors. PI3K class IA is composed of a catalytic isoform encoded by PI3KCA, PI3KCB, or PI3KCD genes and a regulator subunit; p85α, p85β, or p85γ [15]. PI3K is allosterically activated by extracellular signals including growth factors, hormones, or cytokines binding to cognate tyrosine kinase receptors (TKR), promoting its dimerization and phosphorylation [16]. Protein kinase B (PKB), also known as Akt, are serine/threonine protein kinases with three isoforms (Akt1, Akt2 and Akt3) involved in apoptosis, cell proliferation, lysosomal biogenesis, autophagy, cell migration, angiogenesis, and glucose metabolism. One of the activities of Akt is activating the mechanistic target of rapamycin (mTOR), a serine/threonine kinase, which occurs in two different complexes: mTOR complex 1 (mTORC1) and mTOR complex 2 (mTORC2). Akt is recruited to the cell membrane by PI3K, phosphorylated, and activated by phosphoinositide dependent kinases (PDKs). However, for complete Akt activation, it is phosphorylated by mTORC2 in the position Ser473 [17]. The function of mTORC1 is related to the cell metabolism, controlling protein synthesis and autophagy and promoting cell growth when energy is available. mTORC2 is involved in cell proliferation control, cytoskeletal structure, and survival. Importantly, the tumor suppressor gene phosphatase and tensin homolog (PTEN) negatively regulate the anti-apoptotic PI3K/Akt/mTOR signaling pathway activation, thus reducing reactive oxygen species (ROS) in cells [18].

### 2.2. PI3K/Akt/mTOR Activation in Cancer

PI3K/Akt/mTOR pathway has been shown to be altered in different human cancers. For instance, patients with hepatocellular carcinoma (HCC) showed different survival associated with altered PI3K/AKT/mTOR signaling. Indeed, eight to twenty-eight subtype-specific genes have pivotal roles in controlling the metabolic network, with a possible use in therapy [19]. On the other hand, the PI3K/Akt/mTOR pathway has been involved in breast cancer initiation and progression with promising drugs designed for targeting this signaling pathway [20]. Interestingly, PI3K/Akt/mTOR has been associated with the pro-viral-integration site for Moloney-murine leukemia virus (PIM) kinases, with a critical role in the regulation of different proteins involved in ovarian tumorigenesis [21]. Moreover, this pathway is dysregulated and can affect cell transformation, tumor progression, cell survival, and drug resistance in colorectal cancer patients [22]. Additionally, PI3K/Akt/mTOR signaling is dysregulated in renal cell carcinomas (RCCs), and mTOR inhibitors such as temsirolimus and everolimus are important therapeutic options for RCC treatment [23]. Everolimus works similarly to Rapamycin as an inhibitor. Phase 2 clinical trials indicated that everolimus had limited activity, and another clinical trial reported that the treatment displayed significant HNSCC antitumor effects [24]. Of note, the deregulation of PI3K/Akt/mTOR and MAPK signaling pathways can suggest an active involvement in endometrial carcinomas [25]. In pancreatic neuroendocrine tumors (pNETs), this pathway is often deregulated and mTOR inhibitors have demonstrated efficacy for treatment [26]. Interestingly, the tumor suppressive miR-99b-5p has been involved in regulating PI3K/AKT/mTOR activity in prostate cancer (PCa). Indeed, a significant reduction in nuclear mTOR and androgen receptor (AR) has been detected, targeting the AR-mTOR axis using miR-99b-5p. This finding suggests that miR-99b-5p is a novel potential therapeutic target in this type of cancer [27]. Other cancers such as glioblastoma [28], lung [29], multiple myeloma [30] and lymphomas [31] are additionally characterized by PI3K/Akt/mTOR alterations.

## 3. Human Papillomavirus in Head and Neck Carcinogenesis

### 3.1. HPV Structure and Tropism

Human papillomaviruses (HPVs) belong to the papillomaviridae family of non-enveloped viruses [32]. The viral particle is composed of a capsid with icosahedral symmetry harboring 72 pentameric capsomeres, and a circular double-stranded DNA genome containing approximately 8000 base pairs (bp) [33]. Its genome is organized into three regions: (1) the early region (Region E), which encodes for non-structural proteins (E1–E7) involved in viral replication; (2) the late region that contains the L1 and L2 open reading frames (ORFs), which encode for major and minor structural proteins; and (3) the non-coding regulatory region, also called the long control region (LCR). The latter, with an approximate length of ~1000 bp (12% of the viral genome), is divided into three segments: the 5′ or distal segment contains transcription termination signals and a nuclear matrix binding region; the central segment, which contains an epithelial cell-specific enhancer with binding sites for cellular or viral transcription factors that control transcription and viral genome replication; and finally, the 3′ segment, which contains the origin of replication and the early promoter [34,35]. To date, around 210 HPV genotypes have been characterized, which are classified into five genera (α, β, γ, μ, ν) based on the analysis of their genome sequence and tropism [36,37]. Alpha (α)-papillomaviruses, with a mucosal or cutaneous tropism, have been the best-characterized group due to their involvement in cancer [32]. According to their oncogenic potential, α-papillomaviruses have been divided into low- (LR) and high-risk (HR) of cancer. LR-HPV genotypes, HPV6 and 11, mainly, are associated with most benign mucosal lesions and condyloma acuminata. In addition, these HPV genotypes are involved in the development of diseases of the upper aerodigestive tract, such as recurrent respiratory papillomatosis [38]. On the other hand, 12 HR-HPV genotypes (16, 18, 31, 33, 35, 39, 45, 51, 52, 56, 58, and 59) have been described as carcinogenic in humans according to the International Agency for Research on Cancer (IARC) [33,36]. HPV shows an exclusive tropism for cutaneous or mucosal epithelia, requiring microlesions for viral entry to susceptible cells in the basal strata. HPV genomes are replicated and amplified in basal strata with expression of early proteins including E1, E2, E5, E6, and E7. Late viral expression occurs in the upper strata in a cell differentiation-dependent manner. Thus, L1 and L2 genes are expressed from the HPV late promoter, leading to viral genome encapsidation, maturation, and viral releasing.

### 3.2. HPV’s Role in Oropharyngeal Cancers

Different epidemiological studies have shown increased HNC prevalence associated with HR-HPV, particularly those cases located in the oropharynx. Indeed, HPV-positive oropharyngeal cancers (OPC) show high prevalence in the US and Europe (~65%) [39,40]. On the other hand, concomitant to this observation, tobacco-smoke-associated HNCs have decreased in the world [8,41]. Importantly, HPV-positive OPC have a different epidemiological, molecular, and clinical profile, characterized by a better prognosis when compared to tobacco-smoke-associated cancers [41,42]. However, the natural course of HPV-positive OPC has not been fully clarified. Indeed, the naturally discontinuous basal membrane in the tonsil crypts may potentially favor HPV infection, which has been suggested to be a reservoir [43]. Additionally, the lymphoid tissue of this area could favor HPV persistence by evading immune surveillance, due to the expression of PD-L1 to suppress the T cell response [43,44]. Importantly, HPV16 and HPV18 account for ~70% of cervical carcinoma cases, while HPV16 alone accounts for over 90% of HPV-positive OPCs [43]. Importantly, even though tobacco smoke is a recognized cofactor in cervical carcinogenesis, such a relationship has not been established in HPV-driven HNCs. Interestingly, while tobacco-associated HNSCCs show a high frequency of TP53 gene mutations [45], HPV-positive OPCs retain the wild-type TP53 gene [46]. The hallmark of HPV-driven carcinogenesis is the overexpression of E6 and E7 oncoproteins [32]. In fact, the HR-HPV E6 oncoprotein recruits the E6-associated protein (E6AP), a ubiquitin ligase that triggers both p53 proteasomal degradation and p21 inhibition, deregulating the G1/S and G2/M checkpoints typically induced by DNA damage, which allows the induction of genomic instability and the accumulation of gene mutations [47,48]. Furthermore, E6 induces the expression and activity of hTERT, the catalytic subunit of telomerase, favoring cell immortalization and proliferation [46,47,48]. On the other hand, HR-HPV E7 promotes cell transformation through the degradation of retinoblastoma (Rb) family members, releasing the transcriptional factor E2F, and the subsequent stimulation of the S phase entry, also promoting p16 overexpression, whose detection is widely used as a surrogate biomarker of oncogenic HPV in clinical specimens [49]. Furthermore, E7 induces cell proliferation by promoting the deregulation of cyclin-dependent kinase inhibitors, p21 and p27, allowing the activation of CDK2 [46]. Although the presence of precancerous lesions has not been established in HPV-positive OPC as occurring in the cervix [50], it is suggested that random viral DNA integration occurs less frequently than in cervical cancer [51,52,53]. Evidence shows that HPV genome integration in HNC cell lines results in viral–host DNA concatemers and E6/E7 amplification, leading to genomic instability and oncogenesis [54]. Additionally, integration was shown to be associated with alterations in the DNA copy number, mRNA transcript abundance and splicing, and both inter- and intrachromosomal rearrangements in primary head and neck cancers [55]. Regarding innate immune signaling, E6 and E7 can modulate the IFN-I signaling pathway. Indeed, HPV E6 binds interferon regulatory factor 3 (IRF3), preventing its transactivation ability and promoting its degradation. This alteration leads to a decreased JAK-STAT pathway response and the disturbance of the IFN-α pathway [56]. HPV E7 appears to be mostly responsible for pattern recognition receptors (PRR) suppression [57], because E7 targets STING for degradation via autophagy by hijacking the PRR component NLRX1 [58]. Additionally, TAP-1, IFN-β, and MCP-1 genes were less expressed in cervical tissue derived from a transgenic mouse with cervical dysplasia, expressing E6/E7 when compared to normal tissues. These findings show the effect of E7 transgene expression in the inactivation of the IRF-1 function in vivo [59].

## 4. PI3K/Akt/mTOR Pathway Activation in HPV-Positive HNCs

### 4.1. PI3K/Akt/mTOR Activation: Epidemiological Aspects

The PI3K/Akt/mTOR signaling pathway is highly activated (~90%) in both HPV-positive and HPV-negative HNCs, with certain differences between SCCs from the oropharynx and the oral cavity [60,61]. Furthermore, genomic studies have revealed that PI3K is the leading disrupted signaling pathway in HPV-positive HNCs, including alterations by mutation or copy number variation (CNV) such as PIK3CA, tumor suppressor gene phosphatase and tensin homolog (PTEN), Akt1, mTOR, among others [62,63]. Although this signaling pathway is robustly activated in HNCs, PTEN expression is related to a favorable survival outcome in tonsillar cancer patients [64]. Additionally, PI3K is equally overexpressed in both HPV-positive smoker and non-smoker patients, with a non-statistically significant impact on survival [65,66]. Moreover, recent observations reveal that PI3K, Akt, and mTOR activity are not prospective molecular markers correlated with improved outcomes (relapse-free survival and overall survival) [67]. A meta-analysis showed that the mutations prevalence was 13% for PIK3CA (95% confidence interval [CI] = 11–14; I2 = 82%; *p* < 0.0001), 4% for PTEN (95% CI = 3–5; I2 = 55%; *p* < 0.0001), 3% for MTOR (95% CI = 2–4; I2 = 5%; *p* = 0.40), and 2% for AKT (95% CI = 1–2; I2 = 50%; *p* = 0.0001) in HNCs worldwide [68]. In particular, a high frequency of PIK3CA and PTEN gene mutations in HPV-positive HNCs have been reported, with a significant association between PIK3CA, PTEN, and tumor HPV status [69,70,71]. Likewise, novel PIK3CA mutations (R115L; G363A; C971R; R975S) have resulted in HNC cell growth associated with high p-Akt levels, and thus increased PI3K pathway activation [72]. Interestingly, PIK3CA missense somatic mutations, PTEN deletion, and complete PTEN loss have also been identified in HR-HPV-positive parotid tumors [73]. Importantly, a high overall survival has been observed in metastatic HPV-positive oropharyngeal cancer patients, yet there are no associations in terms of disease-specific patients’ survival [74,75]. On the other hand, a recent study estimated a lower disease-free survival in HPV-positive patients with PIK3CA mutations compared to patients with wild-type PIK3CA [76]. An in silico approach using The Cancer Genome Atlas (TCGA) database showed that PI3KC3 transcripts are significantly overexpressed in HPV(p16)-positive HNCs (Figure 1, left). Conversely, a study showed that HPV-positive HNSCCs exhibited lower levels of p-Akt at T308 (*p* = 0.028) than HPV-negative cases [77]. Indeed, Akt1 transcript levels were significantly reduced in HPV(p16)-positive patients when compared to those who were HPV(p16)-negative (Figure 1, middle). In terms of clinical outcomes, it has been reported that high Akt1 expression is related to reduced relapse-free survival in a cohort of 24 patients [67]. Importantly, radioresistance is also a parameter that has been studied. Since tumor hypoxia contributes to worse radiotherapy outcomes in HNCs, recent reports have focused on achieving new therapeutic targets [78,79]. Regarding mTOR, a study demonstrated that mTOR varied widely between tumor cores and it was not associated with HPV status or clinicopathological features, although it showed a positive correlation with pre-treatment FDG uptake, suggesting that the prognostic significance of mTOR expression depends on HPV status [80]. By analyzing the TCGA database, we found no differences in mTOR transcript levels between HPV(p16)-positive and negative HNCs (Figure 1, right).

### 4.2. Mechanisms of PI3K/Akt/mTOR Activation by HPV

HPV modulates PI3K/Akt/mTOR signaling by different mechanisms, and most importantly by E6 and E7 interactions with host partners. These effects of HPV-oncoproteins were first reported in human keratinocytes in which the HPV16 E7 oncoprotein enhanced cell migration in an Akt-dependent manner and, by interaction with pRb family members, in turn increased cell proliferation [81,82]. On the other hand, HPV16 E6 expression increases mTORC1 activity by promoting Akt phosphorylation [83]. Interestingly, a study showed that treatment with rapamycin, a mTOR inhibitor, significantly reduced the tumor size in mice transplanted with HPV-positive oral cells [84]. Later, it was shown that mTOR inhibition prevented SQC development in HPV16 E6/E7 transgenic mice after 7,12-dimethylbenz[a]anthracene (DMBA) treatment, decreasing the DNA damage, suggesting an important role for mTOR in HPV-positive oral cancers [85]. Furthermore, HER3, a well-known tyrosine kinase receptor, has shown a high expression as well as its interaction with PI3K in HPV-positive HNCs [86]. Further studies found that both E6 and E7 regulate HER3 levels via a post-transcriptional mechanism, and subsequently by promoting PI3K/Akt signaling activation [87]. Interestingly, E6/E7 knockdown reduced cell proliferation and decreased PI3K levels and p-Akt in ectopically overexpressing HER3 HPV-positive cells, suggesting a novel mechanism whereby HPV oncoproteins lead to PI3K/Akt pathway activation [87]. Consistent with these findings, E6 and E7 ablation also sensitizes HPV-positive HNC cells to increased apoptosis after PI3K inhibition when compared to E6 and E7 knockdown or inhibitor treatment alone. Of note, both E6 and E7 may confer resistance by overexpressing HER3/Akt in response to the PI3K inhibitor [88]. Notably, a recent study showed that Akt induction represses E6/E7 expression in HPV-positive carcinomas under hypoxic conditions [89]. Hypoxia also induces Akt expression and maintains cell survival in HNCs [90]. Likewise, Akt downregulation may sensitize both HPV-positive and HPV-negative HNC cells to hypoxia, suggesting a continuous crosstalk between HPV oncoproteins and Akt activity in tumor environments [91]. Moreover, other studies have demonstrated that E6 promotes Akt phosphorylation in HNC cells; however, this effect is inhibited by secretory leukocyte protease inhibitor (SLPI), a tumor suppressor which has shown a decreased expression in HPV-positive HNCs [92,93]. More importantly, it was found that exogenous SLPI reserves Akt expression in E6-expressing HNSCC cells, and consequently it induces apoptosis and mitigates tumor invasiveness [92]. On the other hand, a significant inhibition of cell proliferation was found after the treatment of HPV-positive oropharyngeal carcinoma cells with mTOR blockers, which presented a relatively high expression of E2/E4/E5. These results suggest that such early viral proteins may also have a role in mTOR-dependent epithelial carcinogenesis [94]. Interestingly, E6 spliced forms (E6*) play an important role in PI3K signaling. Indeed, E6* proteins differentially modulate the human homologue of Drosophila disc large tumor suppressor protein (hDlg) degradation to rebound the levels of activated PTEN and Akt and increase the expression of p-PI3K, contributing to activate MAPKs and thus promoting cell proliferation [95,96]. Of note, Pleckstrin (PLEK2), a novel cancer progression and invasion-related protein, has been directly involved in hypoxia and metastasis in HNCs [97,98]. Furthermore, PLEK-related hub genes are functionally enriched during HPV infection and PI3K-Akt signaling pathway in HNSCCs [98]. Because PLEK2 may require PI3K activity for actin reorganization and cell spreading [99,100], a potential PLEK/PI3K/Akt carcinogenic mechanism in HPV-positive HNCs could be outlined. Interestingly, Guo et al. found a novel Akt3 splice variant with functional activity in primary HPV-positive OPSCCs. Indeed, the inhibition of this Akt3 variant resulted in significant cell growth reduction. However, Akt3 variant knockdown was not proven to affect p-Akt or canonical Akt1 signaling. Additionally, the presence of the Akt variant did not show a correlation with the lack or existence of PI3K/Akt pathway alterations, concluding an independent mechanism toward carcinogenesis [101]. Taken together, these multiple observations reinforce the alteration of the PI3K/Akt/mTOR pathway as a substantial oncogenic-driven mechanism in HPV-positive HNCs (Figure 2).

## 5. PI3K/AKT/mTOR Pathway Targeting in HPV-Positive HNSCCs

### 5.1. HPV Infection in HNC Treatment

HNC treatment includes traditional surgery, radiotherapy, chemotherapy, targeted therapy, and immunotherapy [102,103]. It has been reported that HPV-positive HNCs show high radiosensitivity and chemosensitivity when compared to HPV-negative counterparts [104,105]. The mechanisms by which this occurs have not been fully elucidated, although impaired DNA damage repair, increased G2/M arrest after irradiation and less hypoxic microenvironments have been suggested for HPV-driven HNCs [106]. Additionally, it has been reported that the expression of a functional p53 can be induced by radiotherapy in HPV-positive OPC cells, leading to cell death and increased sensitivity to radiotherapy, which account at least in part for the better prognosis of these cancers [107,108]. Strikingly, it has been demonstrated that long periods of hypoxia increase Akt levels in HPV-positive HNCs. Therefore, Akt inhibition shows a higher radiosensitizing effect in hypoxic HPV-positive cells than in normoxic cells, suggesting a potential role of the Akt blockade in improving hypoxic tumor radiosensitivity [91]. In normal conditions, while p-Akt expression appears to be high in radioresistant HPV-positive HNC cell xenografts post radiation treatment, a decreased expression is observed in sensitive tumors. These reports together suggest the sustained activity of p-Akt as a radioresistance promoter. Viral proteins can also increase sensitivity to DNA damaging agents, as well as an impaired DNA damage response (DDR) [106]. Interestingly, immunotherapy has been shown to be more effective in recurrent/metastatic (R/M) HPV-positive patients [109]. Additionally, a recent meta-analysis described how patients with HPV-positive tumors treated with immunotherapy had a better tumor response rate and higher overall survival than HPV-negative cases. The most common immunotherapy agents used in clinical trials were the monoclonal antibodies Pembrolizumab, Nivolumab, Durvalumab, Atezolizumab, Motolimod, and Monalizumab [110]. In particular, Pembrolizumab and Nivolumab are immune checkpoint inhibitors against Programmed death receptor-1 (PD-1) that have been approved by the Food and Drug Administration (FDA) for platinum-resistant (second line) R/M HNSCC [111]. In addition, both Durvalumab and Atezolizumab block interactions between PD-L1 and its cognate receptor, and have been used as anti-PD-L1 inhibitors [111]. The overexpression of PD-L1 in tumor cells is related to the amount of IFN-γ secreted by immune cells [112]. In HPV-positive HNCs, Programmed Death-Ligand 1 (PD-L1) is responsible for immune evasion through binding to the programmed death-1 (PD-1) receptor and can be expressed by the cells of the immune system [113]. Using immune competent mouse (iKHP mouse), it was determined that HPV-positive HNC tumors showed a better outcome [114], which could be explained by an increased number of CD8+ T-cells and immune cells with increased infiltration capacity [111]. Other aspects could be their use by HPV in relation to the scape of cell cycle regulations such as T-cell tolerance, immunosuppressive cytokines productions, the negative regulation of interferon gamma and the dysregulation of STAT signaling pathway [115].

### 5.2. Chemotherapy in HPV-Positive HNCs

One of the primary HPV-positive HNSCC treatments recently studied is the use of PI3K inhibitors. Rigosertib (ON 01910.Na) is a non-ATP competitive kinase inhibitor initially observed to disrupt PI3K/Akt pathway activation in hematologic cancers [116]. Similarly, it has been shown that Rigosertib therapy significantly reduces tumor growth in HPV-positive HNSCC-transplanted mice and sensitizes PI3KCA-amplified phenotypes [117]. However, the effect on PI3K pathway modulation remained undetermined until Prasad et al. showed a dose-dependent inhibition in PI3K activity. Interestingly, the combined treatment of Rigosertib with either Cisplatin chemotherapy or radiation substantially increased cell death in HPV-positive tumor cells [118]. Furthermore, monotherapy with pan-PI3K inhibitor Buparlisib (BKM120) successfully reduced cell proliferation in a dose-dependent manner, and effectively promoted cell cycle arrest. Despite the results above, only a small radiosensitizing effect was observed and there were no changes in the DNA damage response in combination with radiotherapy [119]. Another drug candidate was NVP-BEZ235 (BEZ235), a potent dual PI3K/mTOR modulator able to induce apoptosis and increase necrosis in human laryngeal SCC cells [120,121]. As observed in colorectal and glioblastoma cells, BEZ235 pre-treatment may radiosensitize HPV-positive and negative HNSCCs [122,123,124]. This mechanism may be explained by decreased DNA double-strand breaks (DSBs) repair in G1 phase cells by the repression of non-homologous end joining (NHEJ) [124]. Strikingly, HPV-positive tongue SCCs treated with BMK120 or BEZ235 were more sensitive than HPV-negative cells for decreasing cell proliferation. Likewise, a combination of BEZ235 and fibroblast growth factor receptor 3 (FGFR3) inhibitor AZD4547 may allow a concentration reduction of both singular agents [125]. EGFR is an upstream target of the PI3K/AKT/mTOR pathway that is frequently overexpressed in HNSCCs [126,127]. Emerging evidence from pre-clinical studies demonstrates a reduction in PI3K/AKT/mTOR pathway activity by the EGFR blockade with cetuximab in HPV-positive cells [128]. Surprisingly, a suggested mechanism whereby cancer cells may become resistant to EFGR inhibitors is the maintenance of PI3K signaling [129], thus combining EGFR inhibitors with PX-866, a novel PI3K-disabling agent whose treatment has revealed a moderate tumor growth fall in the base of tongue and tonsil carcinomas [130]. In addition, this treatment has been used as a plausible strategy to avoid resistance in epithelial cancers [131,132]. Nonetheless, the cetuximab-PX-866 plan has not shown significant differences in clinical outcomes in samples of nearly half of HPV-positive HNSCC patients [132]. Another scheme proposed has been cetuximab with copanlisib (BAY 80-6946), a highly selective pan-class I PI3K inhibitor with a safe profile in B-cell lymphoma treatment [133,134]. Such dual therapy has shown a better response than each drug alone in HPV-positive HNSCC grafted mice [135]. Additionally, alpelisib (BYL719), a potent and selective PI3Ka inhibitor, presents an in vivo solid tumor efficacy [136]. However, new observations provide a mechanism by which HNSCC may escape the antitumor activity of PI3Kα inhibition. By activating the EGFR/PKC/mTOR axis, HNSCC cells overexpress AXL, a ubiquitous TAM family receptor, and subsequently prevent alpelisib growth suppression [137,138]. Therefore, dual treatment with PI3K and EGFR inhibitors is also a therapeutic strategy to avoid cancer-acquired resistance [137]. Another PI3K inhibitor used is afatinib, which blocks the PI3K/Akt/mTOR pathways and produces decreased HPV E7 expression. Furthermore, afatinib suppressed cell proliferation and survival in HPV-positive HNSCCs [139]. It has also been reported that afatinib is highly effective in reducing HNSCC tumor growth with the use the metformin in vivo. Indeed, metformin inhibits the proliferation of carcinoma cells and induces apoptosis through the activation of AMPK and decreased mTOR activity in HNSCC in vitro and in vivo [140].

In phase I clinical studies, the assessment of the combination of radiotherapy, cetuximab, and alpelisib did not detect imaging-persistent disease in 11 evaluated patients, including 10 HPV-positive HNSCC patients [141]. As mentioned above, FGFR inhibitors may also enhance the PI3K inhibitor’s efficacy [125]. As such, a combination of FGFR inhibitor erdafitinib (JNJ-42756493) and alpelisib, improves proliferation response, suggesting a synergistic effect with, potentially, fewer side effects and resistance development [142]. Interestingly, PI3KCA mutations, such as H1047R, demonstrate an abnormal sensitivity to mTOR/PI3K inhibitor BEZ-235 and a subsequent decrease in cell growth in HPV-positive HNC tumor graft models [143]. Even though targeting the TKR/PI3K/Akt/mTOR pathway is a promising strategy for HNC treatment, some compensatory biological mechanisms can operate, reducing the usefulness of specific inhibitions. For instance, mTOR inhibition can promote negative feedback in which S6K is able to phosphorylate Insulin receptor substrate 1 (IRS1) with the subsequent activation of PI3K and tumor growth [144]. In addition, mTOR inhibition frequently results in Akt upregulation, suggesting that combinations of inhibitors will increase the impact of such treatments [145]. Clinical trials based on HPV positive status and PI3K/Akt/mTOR targeting for HNSCC treatment are summarized in Table 1 and Table 2, respectively.

## 6. Conclusions and Remarks

HNC are a heterogeneous group of cancers including oropharyngeal and oral cancer, among others. A subset of HNC, in particular those arising in the oropharynx, are driven by HR-HPV infection, with HPV16 being the most prevalent HPV genotype. The mechanisms by which HR-HPV promotes carcinogenesis include p53 and pRb downregulation by E6 and E7, respectively.

The PI3K/Akt/mTOR signaling pathway is involved in cell growth and survival, maintaining cell homeostasis. Alterations in this pathway are common in cancer, leading to increased cell proliferation, migration, and invasion. Of note, alterations in this pathway are common in both HPV-negative and positive HNCs. The mechanisms by which HR-HPV promotes PI3K/Akt/mTOR activation have been previously characterized. Indeed, HR-HPV E6 and E7 are important mediators promoting the upregulation of this pathway, although it has been suggested that E2/E4/E5 are involved in those cases harboring episomal forms of HR-HPV. Drugs targeting PI3K/Akt/mTOR pathway activation are being tested in clinical trials for HPV-driven HNC treatment. Targeting this signaling pathway may be an important mechanism to control the tumor properties of HPV-driven HNCs. However, combinations of drugs for HNC treatment are an important alternative to avoid compensatory mechanisms when using specific monotherapies.

## Figures and Tables

**Figure 1 biology-12-00672-f001:**
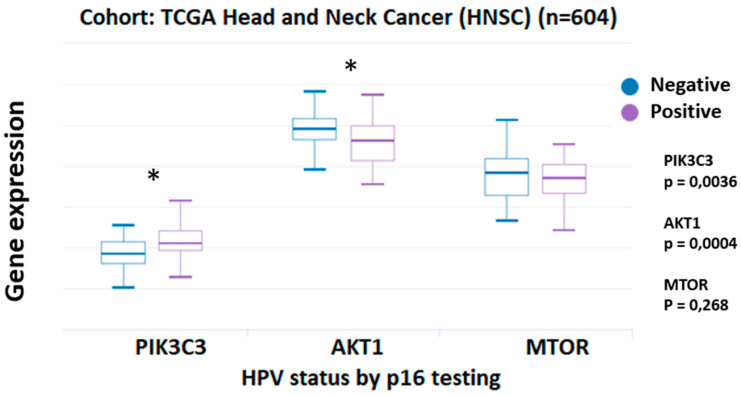
PIK3C3, AKT1 and MTOR transcript expression in HNSCCs (TCGA, n = 604) with HPV positive status using p16 testing. PIK3C3 and AKT showed high expressions in positive HPV samples (*p* = 0.0036 and *p* = 0.0004, respectively). Raw data were extracted from University of California, Santa Cruz, ena.ucsc.edu (Accessed on 14 March 2023), UCSC Xena functional genomics explorer (https://xenabrowser.net). *: *p* < 0.05.

**Figure 2 biology-12-00672-f002:**
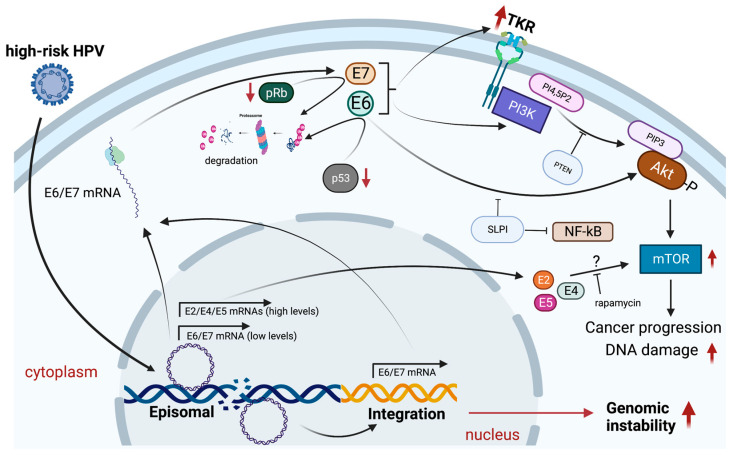
PI3K/AKT/mTOR signaling pathway activation in HPV-positive HNCs. Episomal HPV expresses E2/E4/E5 (high levels) to induce mTOR activation and HNC progression. In addition, episomal and integrated forms of HPV overexpress E6/E7 to promote p53/pRb downregulation and PI3K/Akt/mTOR activation. HR-HPV E6/E7 induce TKR upregulation with E6, promoting Akt phosphorylation and mTOR activation, in turn leading to cancer progression.

**Table 1 biology-12-00672-t001:** Clinical trials based on HPV positive status in NHSCC.

Clinical Trial Number	Project Name	Drug	Target
NCT04252248	Decitabine Treatment in HPV-Induced Anogenital and Head and Neck Cancer Patients After Radiotherapy or as Novel Late Salvage	Dacogen	DNA-demethylating agents
NCT03162224	Safety and Efficacy of MEDI0457 and Durvalumab in Participants with Human Papilloma Virus (HPV) Associated Recurrent/Metastatic Head and Neck Cancer	Durvalumab	Blocking the action of PD-L1
NCT05122221	CRTE7A2-01 TCR-T Cell for HPV-16 Positive Advanced Cervical, Anal, or Head and Neck Cancers	Fludarabine + Cyclophosphamide; interleukin-2	Inhibition of DNA synthesis + Alkylating agents; pro-inflammatory cytokine
NCT05108870	TheraT^®^ Vectors (Vaccines) Combined with Chemotherapy to Treat HPV16 Head and Neck Cancers	Carboplatin; paclitaxel	Binds to EGFR; targets microtubules
NCT04534205	A Clinical Trial Investigating the Safety, Tolerability, and Therapeutic Effects of BNT113 in Combination with Pembrolizumab Versus Pembrolizumab Alone for Patients with a Form of Head and Neck Cancer Positive for Human Papilloma Virus 16 and Expressing the Protein PD-L1	Pembrolizumab	Blocks a protein called PD-1
NCT05286060	Trial of the Combination of GX-188E Vaccination, GX-I7 and Pembrolizumab in Patients with Advanced, Resectable HPV Type 16 and/or 18 Positive Head and Neck Cancer	Pembrolizumab	Blocks a protein called PD-1
NCT05280457	HPV 16-positive and/or HPV 18-positive Recurrent and/or For Patients with Metastatic Head and Neck Cancer to Evaluate GX-188E DNA Vaccination, GX-I7 and Nivolumab Combination Therapy	Nivolumab	Anti-PD1 receptor
NCT05678348	Pyrimethamine as an Inhibitor of NRF2 in HPV-negative Locally Advanced Head and Neck Squamous Cell Carcinoma	Pyrimethamine	Inhibits STAT3 transcriptional activity
NCT01721525	Induction Chemotherapy with Afatinib, Ribavirin, and Weekly Carboplatin/Paclitaxel for Stage IVA/IVB HPV Associated Oropharynx Squamous Cell Cancer (OPSCC)	Afatinib, ribavirin, and carboplatin/paclitaxel	Tyrosine kinase inhibitor; antiviral activity against DNA and RNA viruses; binds to EGFR; targets microtubules
NCT02291055	A Study of ADXS11-001 or MEDI4736 Alone or Combination in Cervical or Human Papillomavirus (HPV)+ Head & Neck Cancer	MEDI4736	Target PD-L1
NCT01084083	Induction Chemotherapy Followed by Cetuximab and Radiation in HPV-Associated Resectable Stage III/IV Oropharynx Cancer	Cetuximab, paclitaxel, cisplatin	Binds to EGFR; targets microtubules; binds to the N7 reactive center on purine residues
NCT03978689	A Phase 1 Study in Patients with HPV16+ Recurrent/Metastatic Head and Neck Squamous Cell Carcinoma	Keytruda	Blocks its interaction with PD-L1 and PD-L2
NCT04260126	Study of PDS0101 and Pembrolizumab Combination I/O in Subjects with HPV16+ Recurrent and/or Metastatic HNSCC	Pembrolizumab	Inhibits PD-1
NCT05541016	Blood-Based Biomarkers to Inform Treatment and Radiation Therapy Decisions for HPV Associated Oropharyngeal Squamous Cell Head and Neck Cancers-DART 2.0	Cisplatin	Binds to the N7 reactive center on purine residues
NCT05582590	Autologous T Cells Targeting HPV16 HPV18 & Survivin in Patients With R/R HPV-related Oropharyngeal Cancers	Fludarabine, cyclophosphamide	Inhibition of DNA synthesis, alkylating agents
NCT03795610	Window of Opportunity Study of IPI-549 in Patients with Locally Advanced HPV+ and HPV− Head and Neck Squamous Cell Carcinoma	IPI-549	Potent inhibitor of PI3K-γ
NCT05357898	Study of SQZ-eAPC-HPV in Patients with HPV16+ Recurrent, Locally Advanced or Metastatic Solid Tumors	Pembrolizumab	Inhibits PD-1

The search was performed at https://clinicaltrials.gov/ (accessed on 10 April 2023) with the keywords “Head and Neck Squamous Cell Carcinoma” AND “HPV”. Clinical trials that also included the action of at least one drug were included.

**Table 2 biology-12-00672-t002:** Clinical trials based on PI3K/Akt/mTOR signaling pathway targeting for HNSCC treatment.

Reference	Project Name	Drug	Target
35667295	Simultaneously targeting ErbB family kinases and PI3K in HPV-positive head and neck squamous cell carcinoma	Afatinib; copanlisib	ErbB kinase inhibitor; FDA-approved PI3K inhibitor
32085396	Dual PI3K/mTOR Inhibitor NVP-BEZ235 Enhances Radiosensitivity of Head and Neck Squamous Cell Carcinoma (HNSCC) Cell Lines Due to Suppressed Double-Strand Break (DSB) Repair by Non-Homologous End Joining	BEZ235	PI3K/Akt/mTOR inhibitor
31292160	Metformin Inhibits Progression of Head and Neck Squamous Cell Carcinoma by Acting Directly on Carcinoma-Initiating Cells	Metformin	Activate AMP-activated protein kinase, and inhibited mTOR signaling both in vitro and in vivo
23730210	Improved clearance during treatment of HPV-positive head and neck cancer through mTOR inhibition	Rapamycin	mTOR inhibitor
32620624	Tyrosine Kinase Inhibitors and Everolimus Reduce IGF1R Expression in HPV16-positive and -negative Squamous Cell Carcinoma	Everolimus	mTOR inhibitor
35790279	Targeted Treatment of HPV16-positive and -negative SCC Cells With Small Molecule Tyrosine Kinase Inhibitors and Everolimus Affects MMP2 and MMP14 Expression	Everolimus	mTOR inhibitor

The search was performed at https://pubmed.ncbi.nlm.nih.gov (accessed on 10 April 2023) with the keywords “Head and Neck Squamous Cell Carcinoma” AND “HPV” AND (“PI3K” OR “AKT” OR “mTOR”). The clinical trial selection was made by reading the full text articles including the terms HPV, HNSCC, and PI3K/AKT/mTOR drug.

## Data Availability

Not applicable.

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
