# Peer review of "PI3K/AKT/mTOR Signaling Pathway in HPV-Driven Head and Neck Carcinogenesis: Therapeutic Implications"

_biology, 2023, doi:10.3390/biology12050672_

Round 1

Reviewer 1 Report

In this review, Perez-Dominguez et al. summarize current understanding of roles of PI3K/Akt/mTOR signaling in driving HNCs and various targeted therapeutic directions aiming at this signaling in treating HPV+ HNCs. Overall, the review is well written with details. However, it is confusing at its present format that if hyperactivation of PI3K is unique to HPV+ HCNs, how HPV infection is poised to modulate PI3K/Akt/mTOR pathway in HCNs, and how HPV infection alters PI3K targeted therapies in HCNs. Please find specific comments below.

Section 2: In addition to E6/E7 function in regulating summarized regulations, roles of E6/E7 in inactivating innate immune signaling and sensing?

As stated in lines 157-158, given hyperactivation of PI3K pathway is observed in both HPV+ and HPV- HNCs, if this hyperactivation is connected with HPV remains unclear and should be further clarified.

Section 3 should be re-organized. 3.4 is the most relevant content to the title but current format is confusing. (1) mouse studies should be separated from cell line or patient studies; (2) details for how HPV oncoproteins interact/regulate certain PI3K members to modulate the pathway activity should be provided (currently it is very correlative and lacks explanations); (3) It will be beneficial to include a figure or table to summarize targets of HPV oncoproteins in PI3K signaling in HNCs.

Section 4 is also confusing at its current format. It may benefit the reader if the authors can organize this section into two sub-sections including (1) how HPV infection affects HCN treatments and (2) how HCN treatments affects HPV-mediated PI3K activation.

Table 1. Will these treatment cause different outcome in HPV- HCNs?

The conclusion is too short that should be further extended to more future directions.

Author Response

Reviewer 1

In this review, Perez-Dominguez et al. summarize current understanding of roles of PI3K/Akt/mTOR signaling in driving HNCs and various targeted therapeutic directions aiming at this signaling in treating HPV+ HNCs. Overall, the review is well written with details. However, it is confusing at its present format that if hyperactivation of PI3K is unique to HPV+ HCNs, how HPV infection is poised to modulate PI3K/Akt/mTOR pathway in HCNs, and how HPV infection alters PI3K targeted therapies in HCNs. Please find specific comments below.

ANSWER. Many thanks for this observation. The manuscript was completely checked for this very important concern.

Section 2: In addition to E6/E7 function in regulating summarized regulations, roles of E6/E7 in inactivating innate immune signaling and sensing?

ANSWER.  The role of E6/E7 in inactivating innate immune signaling was added (page 4)

As stated in lines 157-158, given hyperactivation of PI3K pathway is observed in both HPV+ and HPV- HNCs, if this hyperactivation is connected with HPV remains unclear and should be further clarified.

ANSWER: Additional sentences were added to clarify how PI3K activation is related to HPV-HNCs. The Figure 2 was improved to clarify the involved mechanisms.

Section 3 should be re-organized. 3.4 is the most relevant content to the title but current format is confusing. (1) mouse studies should be separated from cell line or patient studies; (2) details for how HPV oncoproteins interact/regulate certain PI3K members to modulate the pathway activity should be provided (currently it is very correlative and lacks explanations); (3) It will be beneficial to include a figure or table to summarize targets of HPV oncoproteins in PI3K signaling in HNCs.

ANSWER: The manuscript was completely reorganized including first epidemiological aspects, followed by mechanisms of PI3K activation by HPV. A Table with clinical trials for targeting PI3K pathway was included.

Section 4 is also confusing at its current format. It may benefit the reader if the authors can organize this section into two sub-sections including (1) how HPV infection affects HCN treatments and (2) how HCN treatments affects HPV-mediated PI3K activation.

ANSWER: Many thanks for this observation. This section was reorganized in two sections entitled: HPV infection in HNC treatment followed by othe section entitled Chemotherapy in HPV-positive HNCs.

Table 1. Will these treatment cause different outcome in HPV- HCNs?

ANSWER: Previous Table 1 was deleted to include e naew Table 1 including clinical trials targeting PI3K pathway in HPV positive HNCs.

The conclusion is too short that should be further extended to more future directions.

ANSWER: The conclusion was improved, covering the different aspect addressed in this manuscript.

Reviewer 2 Report

Francisco Perez-Dominguez et al., have discussed the role of  PI3K/AKT/mTOR in HR-HPV- 16 associated HNCs development and their impact on therapeutic targets.  In my opinion, the authors should elaborate more on the conclusion section and add one section on the limitation of this therapeutic approach. Especially since the PI3K/AKT/mTOR pathways act in many places, including in the normal condition, how can we target this one exclusively?  The PI3K/AKT/mTOR also cross-talks with many other pathways. 

The authors should also mention the literature search strategy and keywords used for searching the literature.

Author Response

Reviewer 2

Francisco Perez-Dominguez et al., have discussed the role of  PI3K/AKT/mTOR in HR-HPV- 16 associated HNCs development and their impact on therapeutic targets.  In my opinion, the authors should elaborate more on the conclusion section and add one section on the limitation of this therapeutic approach. Especially since the PI3K/AKT/mTOR pathways act in many places, including in the normal condition, how can we target this one exclusively?  The PI3K/AKT/mTOR also cross-talks with many other pathways.

The authors should also mention the literature search strategy and keywords used for searching the literature.

ANSWER: Many thanks for these observations. A sentence including limitations of this therapeutic approach was included, and the keywords used for searching the clinical trials were included in Table 1.

Round 2

Reviewer 1 Report

The authors have addressed my raised concerns.